# Genotype-phenotype characterization and functional reconstitution of pathogenic β-catenin variants from CTNNB1 syndrome patients

Caroline E. Nunes-Xavier[1,2]*, Mercè Pallarès-Sastre[3], Ana Rodríguez-Ramos[1,2], Sonia Bañuelos[4,5], Irune Cortajarena[4,5], Fabio Cavaliere[6,7], Cynthia Ruiz-Espinoza[1,8], Isabel Llano-Rivas[1,9], Maitane García[3], Imanol Amayra[3], Rafael Pulido[1,2,10,*]

**1** Biobizkaia Health Research Institute, Barakaldo, Spain, **2** CIBERER, ISCIII, Madrid, Spain, **3** Neuro-e-Motion Research Team, Department of Psychology, Faculty of Health Sciences. University of Deusto, Bilbao, Spain, **4** Department of Biochemistry and Molecular Biology, University of the Basque Country (UPV/EHU), Leioa, Spain, **5** Biofisika Institute (UPV/EHU-CSIC), University of the Basque Country (UPV/EHU), Leioa, Spain, **6** Achucarro Basque Center for Neuroscience, The Basque Biomodels Platform for Human Research (BBioH), Leioa, Spain, **7** CIBERNED, ISCIII, Madrid, Spain, **8** Department of Pediatrics, Basurto University Hospital, Bilbao, Spain, **9** Department of Genetics, Cruces University Hospital, Barakaldo, Spain, **10** Ikerbasque, The Basque Foundation for Science, Bilbao, Spain

\* rpulidomurillo@gmail.com, rafael.pulidomurillo@bio-bizkaia.eus (RP); carolinenunesxavier@gmail.com, carolineenlisabeth.nunes-xavier@bio-bizkaia.eus (CENX)

## Abstract

Germline variants in the *CTNNB1* gene, encoding β-catenin protein, cause severe neurodevelopmental alterations manifested early in the infancy, and define the CTNNB1 syndrome. Patients with CTNNB1 syndrome display heterogeneous clinical manifestations, and most of them carry *CTNNB1* pathogenic nonsense or frameshift variants that generate premature termination codons (PTC). We have previously described the neuropsychological manifestations of a group of CTNNB1 syndrome patients harboring novel β-catenin variants. Here, we have analysed the molecular and functional characterization of these β-catenin variants, performed genotype-phenotype analyses, and tested for β-catenin functional reconstitution. We describe a complex variety of N-terminal and C-terminal truncated β-catenin proteoforms generated by PTC. Protein stability of truncated proteoforms was variable, as indicated by their expression levels and biophysical analysis, and high protein stability correlated with better patient performance in visuospatial tests. Transcriptional activity was abrogated in most of the β-catenin variants, although some specific truncations, as well as a three-residues in-frame deletion variant, retained partial transcriptional activity. Reconstitution of full-length β-catenin expression and function was achieved in specific β-catenin PTC variants by induction of translational readthrough with aminoglycosides and protein synthesis stimulators. Inhibition of β-catenin degradation by MG-132 proteasome inhibitor also resulted in partial rescue of β-catenin transcriptional activity. Our results suggest the existence of intricate patterns of truncated β-catenin proteoforms in CTNNB1 syndrome patients, which may correlate with

**Data availability statement:** All relevant data are within the paper. The submission contains all raw data required to replicate the results of the study.

**Funding:** This work has been funded by Fundación FEDER, Spain (AI-2023-017) to RP; Fundación Inocente Inocente, Spain (FII2024-69) to RP; CIBERER, ISCIII, Spain to RP; and the Basque Government, Spain (IT1454-22) to SB. CENX is funded by Miguel Servet Research Contract from ISCIII (CP20/00008, Spain and co-funded by European Union). MPS is funded by Ministry of Sciences, Innovation and Universities of Spain (FPU22/00391). ARR is funded by CIBER, ISCIII, Spain. RP is funded by Ikerbasque, The Basque Foundation for Science, Spain. The funders did not play any role in the study design, data collection and analysis, decision to publish, or preparation of the manuscript.

**Competing interests:** The authors have declared that no competing interests exist.

clinical manifestations, and provide insights to increase the function of β-catenin in patients carrying *CTNNB1* pathogenic variants.

## Author summary

CTNNB1 syndrome is a severe rare disease caused by genetic alterations in the *CTNNB1* gene. Children with CTNNB1 syndrome have anomalous neurodevelopment in the early infancy, which causes multiple and heterogenous physical and psychological disabilities. CTNNB1 syndrome does not have a cure today, and patients are only treated with daily care support, physiotherapy, and palliative therapies, depending on the severity of the disease, which is highly variable. *CTNNB1* gene produces the β-catenin protein, which is essential for health, and the *CTNNB1* alterations that cause the disease impair the function of β-catenin. This study aims to characterize the specific molecular alterations in the *CTNNB1*/β-catenin variants present in CTNNB1 syndrome patients, and to investigate the feasibility of novel therapies that help to reconstitute the function of β-catenin, which may alleviate the damage caused by the disease.

## Introduction

CTNNB1 syndrome (neurodevelopmental disorder with progressive spastic diplegia and visual defects; NEDSDV; ORPHA:404473; OMIM: 615075) is a severe neurodevelopmental disorder that manifests during the newborn period and it is caused by a heterozygous variant, in most cases *de novo*, in the *CTNNB1* gene in the chromosome 3p22.1, which results in haploinsufficiency [1–3]. Inheritance is autosomal dominant and the estimated incidence is about 1:30,000 [4]. CTNNB1 syndrome patients experience global developmental delay and they show a great phenotypic variety. The major physical manifestations are microcephaly, abnormal muscle tone and motor problems, visual alterations including strabismus and vitreoretinopathy, and in some cases congenital heart defects. Neuropsychological manifestations are related with autism spectrum disease (ASD)-like features, and include language disorders, learning and cognitive deficits, and behavioral alterations [5–12].

   The *CTNNB1* gene encodes the β-catenin protein, which is made up of 781 amino acids that adopt a rod-like structure with 12 armadillo-type repeat domains flanked by flexible N-terminal and C-terminal regions [13,14]. β-catenin is an essential protein in development, particularly in the nervous system, by virtue of its dual function. On the one hand, it has a central role as a transcription factor in the Wnt/β-catenin signaling pathway, which regulates the balance between proliferation and differentiation in cells, thus directing embryogenesis. On the other hand, β-catenin is important in the morphogenesis of epithelial tissue by forming part of the cell adherens junctions, structural support multiprotein complexes that include β-catenin, α-catenin and E-cadherin, among other molecules. The signaling and scaffolding

role of β-catenin/N-cadherin is also essential in synapse morphogenesis and synaptic plasticity, the bases of memory and learning [15–18]. In addition to its important role in embryonic development, β-catenin is essential for tissue homeostasis in the adult organism, and its deregulation is associated with hyperproliferative pathologies such as cancer, or with neurological diseases including Alzheimer disease, Parkinson disease, and ASD [3,19–21]. β-catenin function is regulated at multiple levels, and a major regulatory mechanism involves its active degradation by the proteasome upon sequential phosphorylation at its N-terminus by CKI and GSK3 in the context of the multiprotein destruction complex. Wnt signaling inhibits this phosphorylation, facilitating β-catenin accumulation and translocation to the nucleus, where it exerts its transcription factor activity [21,22]. In this regard, GSK3 inhibitors have been shown to reconstitute β-catenin protein levels in a haploinsufficiency preclinical mouse model of *Ctnnb1+/-*, and ameliorate the CTNNB1 syndrome-related phenotype [23].

Patient germline variants of *CTNNB1* are distributed throughout the gene, with a predominance of nonsense and frameshift variants generating premature termination codons (PTC) in the *CTNNB1* coding sequence. This generates pathogenic β-catenin truncated proteins and β-catenin mRNAs potentially prone to degradation by nonsense-mediated mRNA decay (NMD) transcript surveillance pathway [24–26]. Of interest for therapy, biosynthesis of full-length proteins from PTC variants can be pharmacologically reconstituted by translational readthrough, which incorporates an amino acid at the position of the PTC enabling proper translation elongation [27–29]. Missense or splice-site variants, as well as partial or total *CTNNB1* gene deletions, have been documented at a lesser extent in patients [2,9,30]. The most prevalent *CTNNB1* variants would give rise to truncated variants of the protein, which could display decreased stability and accelerated degradation or manifest altered functions [31]. In any case, it is believed that germline variants in *CTNNB1* would result in a deficit of β-catenin expression or function in cells. However, the specific consequences of the different variants on the functionality of β-catenin are mostly unknown.

We have previously reported the heterogeneous clinical and neuropsychological manifestations of a cohort of Spanish *CTNNB1* syndrome patients harboring novel β-catenin variants, including a variety of nonsense and frameshift variants [32,33]. Here, we have analysed at biochemical and functional level the properties of these β-catenin variants, tested their reconstitution by inhibitors of the proteasome and translational readthrough inducers, and carried out genotype-phenotype analyses. Our results illustrate the existence of distinct patterns of protein stability associated with specific β-catenin truncated proteoforms, and suggest the possibility of correlating the protein stability of β-catenin variants with phenotypic characteristics of CTNNB1 syndrome patients. We also show that specific β-catenin PTC variants can be functionally reconstituted by translational readthrough, suggesting the possibility of novel therapeutic interventions for CTNNB1 syndrome patients.

## Results

### Biochemical characterization of β-catenin proteoforms translated from pathogenic CTNNB1 variants

We have clinically characterized an extensive group of Spanish CTNNB1 syndrome patients displaying heterogeneous clinical phenotypes, and carrying in most of the cases nonsense or frameshift germline variants at *CTNNB1* gene [32,33]. Here, we have investigated the molecular and functional properties of the β-catenin variants potentially present in these patients, including 18 variants containing premature termination codons (PTC) (8 nonsense variants and 10 frameshift variants) and 1 variant containing a 3-amino acid in-frame deletion (Fig 1 and Table 1). The 3 splicing variants and 2 gene deletions from our CTNNB1 syndrome cohort were not experimentally analysed (Table 1).

Variants were generated *in vitro* in a human *CTNNB1* cDNA cloned into a mammalian expression plasmid and proteins were ectopically expressed in mammalian cells by transient transfection. Expression of the β-catenin variants was monitored by immunoblot using anti-β-catenin N-terminal and C-terminal antibodies (Fig 2A). As shown, immunoblot with anti-β-catenin N-terminal antibody revealed the presence of β-catenin C-terminal truncated proteoforms of variable size,

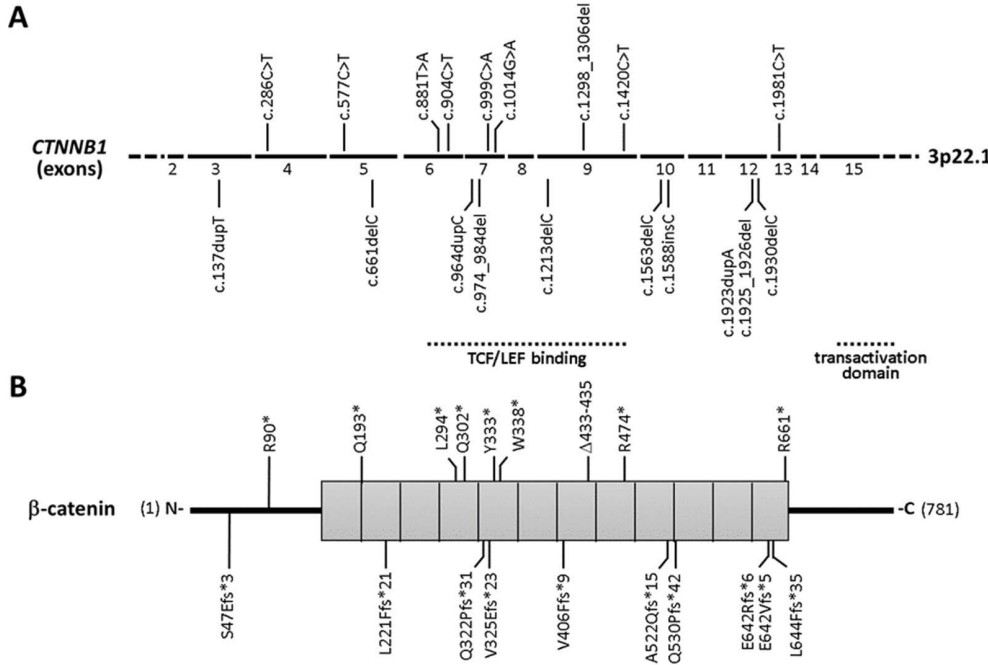

**Fig 1. Schematic depiction of the *CTNNB1* variants analysed in this study. (A)** *CTNNB1* gene is depicted, with indication of the coding exons, from 2 to 15. In the top, nucleotide substitutions causing premature termination codons (PTC) and an in-frame deletion of 9 nucleotides are denoted. In the bottom, frameshift variants causing PTC are denoted. The variants are indicated with nucleotide numbering (NM_001904), following the HGVS recommended nomenclature. **(B)**β-catenin protein is depicted, from amino acid 1 to 781 (NP_001895). Squares correspond to armadillo domains. Variants are indicated as in **(A)**, using the one-letter amino acid code. The protein regions directly involved in transcriptional activity are indicated.

which, in some cases, were expressed at levels comparable with β-catenin wild type. Immunoblot with anti-β-catenin C-terminal antibody showed the translation of β-catenin N-terminal truncated proteins when the PTC was within the first 200 residues of β-catenin amino acid sequence (variants R90*, Q193*, and S47Efs*3). These N-terminal truncations are likely generated by the initiation of translation at methionine downstream to the PTC. The 3-amino acid in-frame deletion variant (Δ433–435) migrated and was expressed similarly than β-catenin wild type. The relative expression levels of the distinct β-catenin variants are illustrated in **S1 Fig**. Thus, a variety of β-catenin N-terminal and C-terminal truncated proteoforms are translated *in vitro* in the presence of *CTNNB1* PTC variants present in CTNNB1 syndrome patients (depicted in Fig 2B).

### Transcriptional activity of β-catenin pathogenic variants

Next, we evaluated the function of the β-catenin pathogenic variants, ectopically expressed in mammalian cells, using a TCF/LEF luciferase reporter assay, which informs on the transcriptional activity of β-catenin (Fig 2C). Most of the variants displayed a complete loss of transcriptional activity. Interestingly, some β-catenin variants, including R90*, R474*, and S47Efs*3, consistently displayed weak transcriptional activity. The Δ433–435 variant displayed about 50% of transcriptional activity, when compared with β-catenin wild type. These findings demonstrate loss-of-function on most of the β-catenin variants, and suggest that specific β-catenin truncated variants, potentially present in CTNNB1 syndrome patients, retain partial transcriptional activity.

To gain further information on the effect of N-terminal and C-terminal truncations on the function of β-catenin, we generated N-terminal truncation variants starting at the methionine residues downstream to the most N-terminal PTC present

**Table 1. CTNNB1 pathogenic variants analysed in the study.**

| DNA variant | Protein variant | Exon (Intron) | Patient number | Sex | Age |
|---|---|---|---|---|---|
| *Nonsense variants* | | | | | |
| c.268C>T | p.(Arg90Ter)/R90*(TGA) | 4 | 23 | M | 8y 9m |
| c.577C>T | p.(Gln193Ter)/Q193*(TAG) | 5 | 3 | F | 7y 8m |
| c.881T>A | p.(Leu294Ter)/L294*(TAG) | 6 | 6 | F | 5y 11m |
| c.904C>T | p.(Gln302Ter)/Q302*(TAA) | 6 | 25 | F | 6y 8m |
| c.999C>A | p.(Tyr333Ter)/Y333*(TAA) | 7 | 11 | M | 4y 1 m |
| c.1014G>A | p.(Trp338Ter)/W338*(TGA) | 7 | 19 | F | 3y 11m |
| c.1420C>T | p.(Arg474Ter)/R474*(TGA) | 9 | 7 | M | 10y 9m |
| c.1981C>T | p.(Arg661Ter)/R661*(TGA) | 13 | 1 | F | 13y |
| *Frameshift variants* | | | | | |
| c.137dupT | p.(Ser47GlufsTer3)/ S47Efs*3 | 3 | 14 | F | 8y 10m |
| | | | 21 | F | 15y 7m |
| c.661delC | p.(Leu221PhefsTer21)/ L221Ffs*21 | 5 | 20 | F | 2y 4m |
| c.964dupC | p.(Gln322ProfsTer31)/ Q322Pfs*31 | 7 | 12 | F | 4y 10m |
| c.974_984del | p.(Val325GlufsTer23)/ V325Efs*23 | 7 | 8 | M | 6y 11m |
| c.1213delC | p.(Val406PhefsTer9)/ V406Ffs*9 | 9 | 4 | F | 8y 5m |
| c.1563delC | p.(Ala522GlnfsTer15)/ A522Qfs*15 | 10 | 10 | M | 15y 1m |
| c.1588insC | p.(Gln530ProfsTer42)/ Q530Pfs*42 | 10 | 9 | M | 10y 5m |
| c.1923dupA | p.(Glu642ArgfsTer6)/ E642Rfs*6 | 12 | 5 | F | 5y 9m |
| c.1925_1926del | p.(Glu642ValfsTer5)/ E642Vfs*5 | 12 | 22 | M | 7y 4m |
| c.1930delC | p.(Leu644PhefsTer35)/ L644Ffs*35 | 12 | 15 | F | 17y |
| *In-frame deletion* | | | | | |
| c.1298_1306del | p.(Lys433_Lys435del)/ Δ433-435 | 9 | 18 | F | 2y 11m |
| *Splicing variants* | | | | | |
| c.1082-1G>C | | (7) | 17 | F | 4y 6m |
| c.2076+1G>A | | (13) | 24 | M | 4y 9m |
| c.2137+2T>C | | (14) | 16 | M | 6y |
| *Gene deletions* | | | | | |
| c.(936+1_937-1)_(*1106_?)del | | (6–15?) | 13 | M | 4y 4m |
| del 3p22.1 | | | 2 | F | 7y 8m |

Variants are indicated following HGVS recommended nomenclature. Protein variants are also indicated using the single-letter amino acid code. In the case of nonsense variants, the identity of the PTC is indicated. The patient's age was recorded at the time of cognitive assessment.

in the patients from our cohort, namely M88, M98, and M194 β-catenin variants. In addition, we generated C-terminal truncation variants lacking the amino acids downstream to PTC Y333*, W338*, R474*, and R661*, namely 1–332, 1–337, 1–473, and 1–660 β-catenin variants. All proteoforms migrated in SDS-PAGE gels and displayed anti-β-catenin antibody

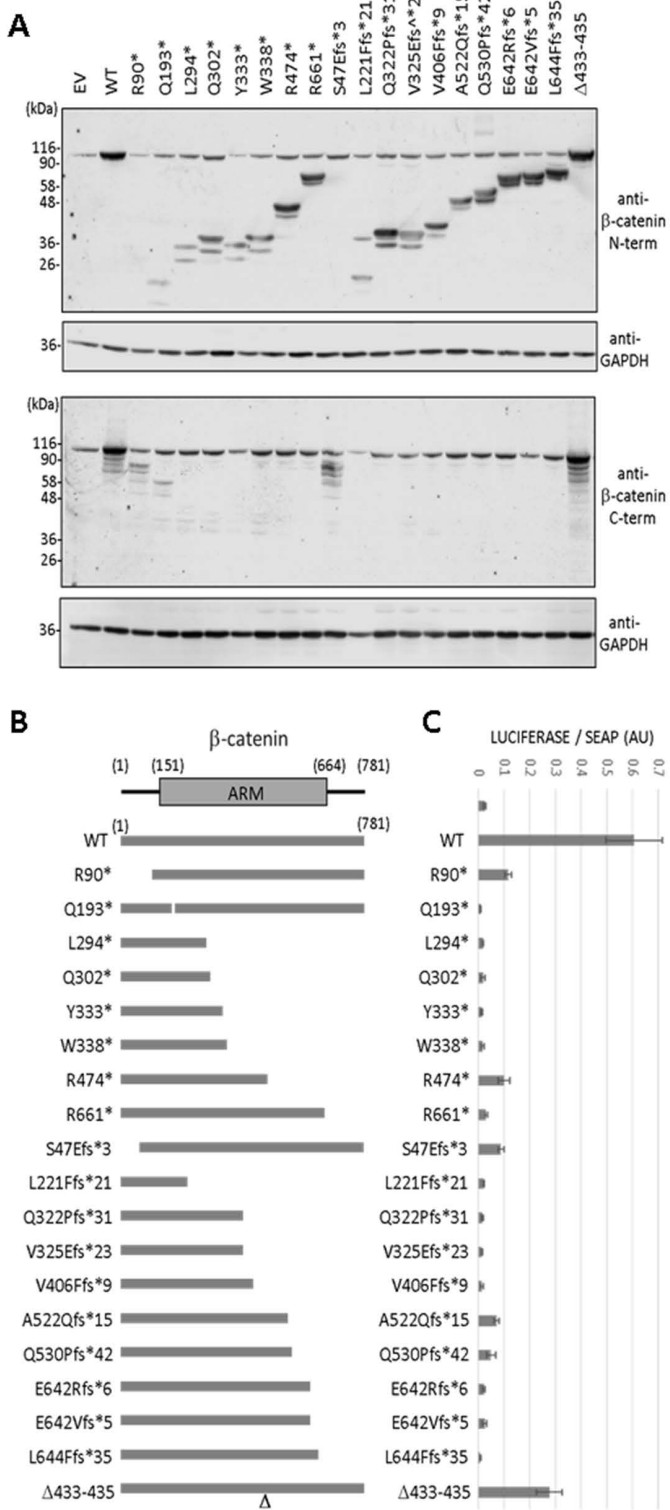

**Fig 2. (A) Expression of β-catenin variants analysed in this study.** COS-7 cells were transfected with pRK5 mammalian expression plasmid encoding the indicated β-catenin variants (EV, empty vector; WT, wild type), and cell lysates were analysed by immunoblot with anti-β-catenin N-terminal (N-term) or C-terminal (C-term) antibodies, as indicated. Immunoblots with anti-GAPDH are shown as controls. **(B)** Schematic representation

of the β-catenin proteoforms generated by the β-catenin variants analysed in this study. Lines indicate the relative length of the different proteoforms. **(C)** Transcriptional activity of β-catenin variants. COS-7 cells were transfected with plasmids encoding the indicated β-catenin variants (EV, empty vector; WT, wild type), and processed for TCF/LEF-driven luciferase activity. Luminescence is shown in arbitrary units (AU), from at least two independent experiments (SEAP, secreted alkaline phosphatase).

reactivity according to their amino acid composition (Fig 3A). β-catenin N-terminal variants displayed diminished transcriptional activity, with M88 and M98 variants consistently showing higher activity than the others (Fig 3B). β-catenin C-terminal variants also displayed diminished transcriptional activity, with 1–473 variant consistently showing more activity than the others (Fig 3B). Additional experiments were performed using β-catenin compound mutations generating C-terminal truncations together with the pathogenic amino acid substitution L388P, which has been reported to reduce β-catenin transcriptional activity [30] (Fig 3C–D). As shown, the compound mutations resulted in full abrogation of β-catenin transcriptional activity.

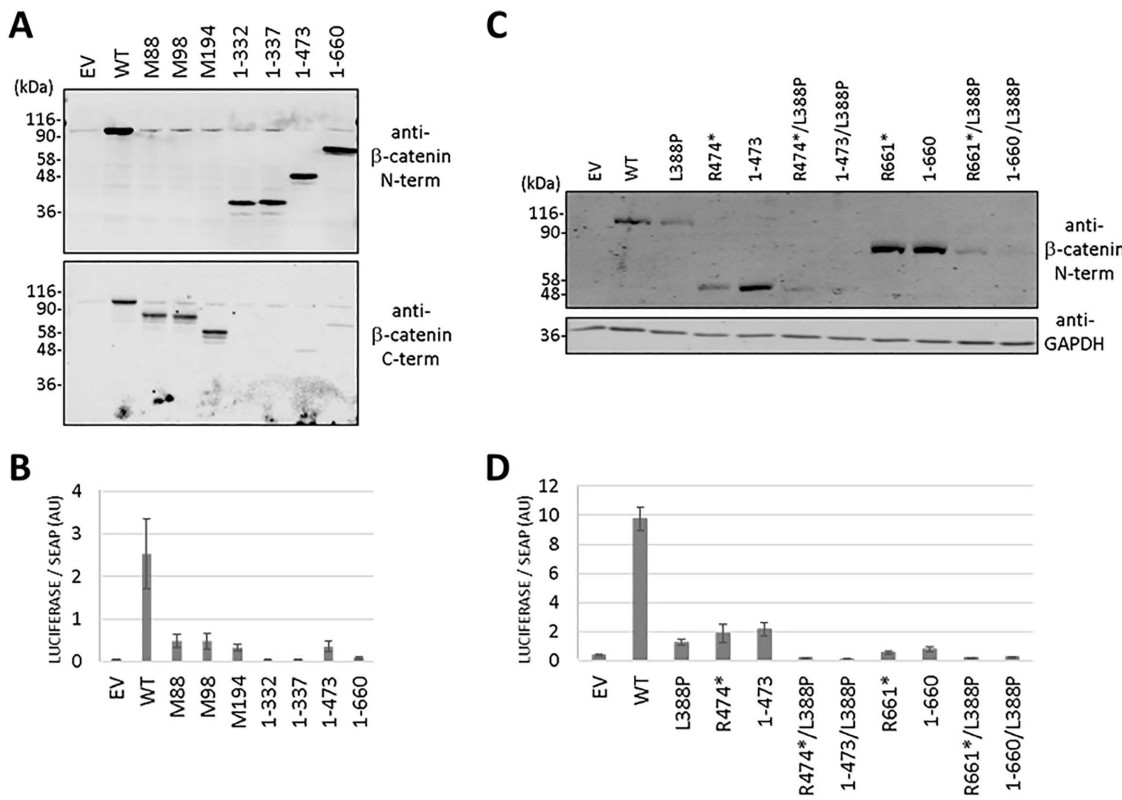

**Fig 3. (A) Expression of N-terminal and C-terminal truncated β-catenin proteoforms.** COS-7 cells were transfected with plasmids encoding β-catenin proteoforms starting at methionine 88 (M88), methionine 98 (M98) or methionine 194 (M194), or starting at methionine 1 and extending to amino acid 332 (1-332), amino acid 337 (1-337), amino acid 473 (1-473) or amino acid 660 (1-660), as indicated. Cell lysates were processed for immunoblot with anti-β-catenin N-terminal (N-term) or C-terminal (C-term) antibodies, as indicated. **(B)** Transcriptional activity of β-catenin proteoforms. COS-7 cells were transfected with plasmids encoding the indicated β-catenin proteoforms (EV, empty vector; WT, wild type; M88, 88-781; M98 98-781; M194, 194-781), and processed for TCF/LEF-driven luciferase activity. Luminescence is shown in arbitrary units (AU), from at least two independent experiments. **(C)** Expression of the β-catenin single- and compound-mutation variants analysed in **(D)**. COS-7 cells were transfected with plasmids encoding the indicated β-catenin variants, and cell lysates were processed for immunoblot with anti-β-catenin N-terminal antibody. **(D)** Transcriptional activity of β-catenin single- and compound-mutation variants. Cells were processed as in **(B)** (EV, empty vector; WT, wild type), from at least two independent experiments.

These results illustrate that both N-terminal and C-terminal β-catenin amino acids are important for transcriptional activity. Importantly, M88, M98, and 1–473 β-catenin proteoforms retained partial transcriptional activity, in concordance with the activity retained by R90*, S47Efs*3, and R474* β-catenin variants. This supports the possibility that specific β-catenin truncated variants potentially present in patients display partial transcriptional activity.

## Protein stability of β-catenin pathogenic variants

Most of the β-catenin PTC pathogenic variants studied generated truncated proteins with variable expression levels. Next, we investigated the protein stability of these β-catenin truncated proteoforms analyzing their expression in transfected cells in the presence of the protein synthesis inhibitor cycloheximide. In general, the shorter β-catenin truncated proteoforms displayed compromised protein stability, when compared with β-catenin wild type, whereas the longer β-catenin truncated proteoforms showed stability similar to β-catenin wild type (Fig 4A). A clustering of patients based on protein stability (H, high stability; L, low stability) of their β-catenin variants is shown in S1 Table. Since most of the differences in protein stability could be observed after the R474 residue, we performed more detailed experiments comparing the R474* and R661* β-catenin variants. R474* variant protein expression was greatly diminished upon cycloheximide cell treatment, when compared with β-catenin wild type, suggesting a very rapid turn-over indicative of diminished protein stability. On the other hand, stability of R661* variant was similar than of β-catenin wild type (Fig 4B).

To investigate the possibility that R474* and R661* β-catenin variants present different stability due to different conformational status, the two variants as well as the wild type β-catenin were overexpressed and purified from bacteria, and analysed by circular dichroism (CD) spectroscopy (Fig 4C). As shown, R474* and R661* β-catenin variants displayed far UV CD spectra similar to that of wild type β-catenin, and typical of mostly α-helical proteins, reflecting that although truncated they can adopt a native-like secondary structure [14]. However, the ellipticity at 208 and 222 nm of the R474* and R661* variants was significantly decreased as compared to wild type β-catenin, which indicates lower content of α-helix. The fact that the ellipticity drop is more evident/stronger in the spectrum of R474* suggests that this variant is less structured than R661*, in agreement with their relative stability in cells. We also performed cycloheximide time-course experiments with the R90* and the S47Efs*3 reinitiation of translation variants, and with the S33C pro-oncogenic variant [34] (Fig 4D). The R90*, S47Efs* and S33C variants displayed similar degradation rate as β-catenin wild type. Together, these findings suggest the existence of specific patterns of protein stability for the distinct β-catenin truncated variants.

## Genotype-phenotype analyses in CTNNB1 syndrome patients

Our molecular studies on the β-catenin variants allowed us to discriminate several groups of variants based on protein stability, reinitiation of translation, or residual transcriptional activity (S1 Table). We performed inter-group analyses to investigate potential differences in these patient groups with their clinical and psychological manifestations (S2 Table), which we have previously described [32,33]. No statistically significant differences were found when the reinitiation of translation or the residual transcriptional activity groups were analysed, with the exception of the Total Vineland-3 (ABC) assessment (S2 Table), which showed better performance for the group with non-residual transcriptional activity ($n = 18$; $M = 65.78$; $M$age $= 6.28$ years old) compared to the residual transcriptional activity group 1 ($n = 7$; $M = 52.86$; $M$age $= 9.78$ years old). However, given that sample size and age are notoriously different between groups, this result should be interpreted with caution. In addition, we found statistically significant differences between the patient groups defined by protein stability in the visuospatial integration and functioning and object assembly performance, with the group of patients displaying high protein stability (H) showing better performance than patients with low protein stability (L) in both tests. Specifically, differences were found among groups in the Matrices test ($U = 27$; $p = 0.009$) and in the Object Assembly test ($U = 25.5$; $p = 0.006$) from the Wechsler Nonverbal Scale of Ability (Fig 5A). Metric multidimensional analysis confirmed the similarities and differences between the subjects ($RSQ = 1.000$; $S$-stress $= 0.000$) (Fig 5B). Statistical differences were also

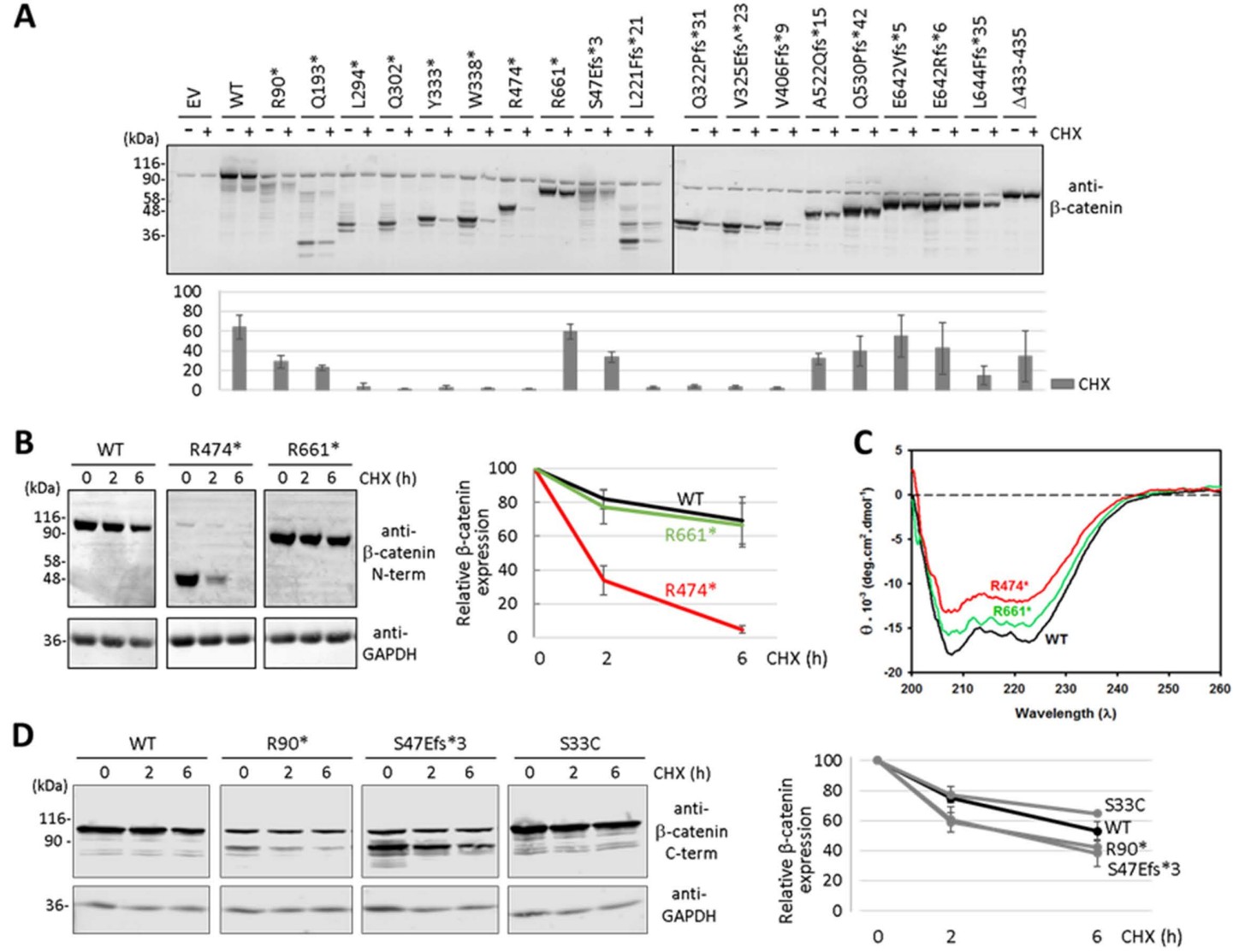

**Fig 4. Stability of β-catenin variants. (A)** COS-7 cells were transfected with plasmids encoding the β-catenin variants (EV, empty vector; WT, wild type) and kept untreated (-) or incubated (+) in the presence of cycloheximide (CHX, 800 μg/ml) for 6 h to monitor protein degradation. The top panel shows a representative immunoblot using sequentially anti-β-catenin N-terminal and C-terminal antibodies. The bottom panel shows the relative quantification of the representative bands from each variant. Bars represent the relative expression after CHX treatment, with regard to the expression under the untreated conditions ± SD, from at least two independent experiments. **(B)** COS-7 cells were processed as in (A) and kept untreated (0) or incubated in the presence of cycloheximide (CHX, 800 μg/ml) for 2h or 6h, as indicated. Immunoblots were performed using anti-β-catenin N-terminal or anti-GAPDH antibodies, as indicated. In the left panel, representative immunoblots are shown. In the right panel, relative quantification of the bands from each variant is shown, from at least two independent experiments. **(C)** Far UV CD spectra of His-ZZ-tagged wild type β-catenin (black) and the variants R474* (red) and R661* (green) at 1 μM in buffer 20 mM Tris/ HCl, 50 mM NaCl, pH 8.0. **(D)** COS-7 cells were processed as in (B) using the indicated variants. In the left panel, representative immunoblots are shown. In the right panel, relative quantification of the bands from each variant is shown, from at least two independent experiments.

found in three subdomains from the Vineland-3, specifically in the Written, Domestic and Community subdomains, which also displayed better functioning for the H group. However, this can be explained by the statistical differences regarding the age between groups ($U = 37.5$; $p = 0.026$). The H stability group patients (*M*age = 9.05 years old) were older than the L

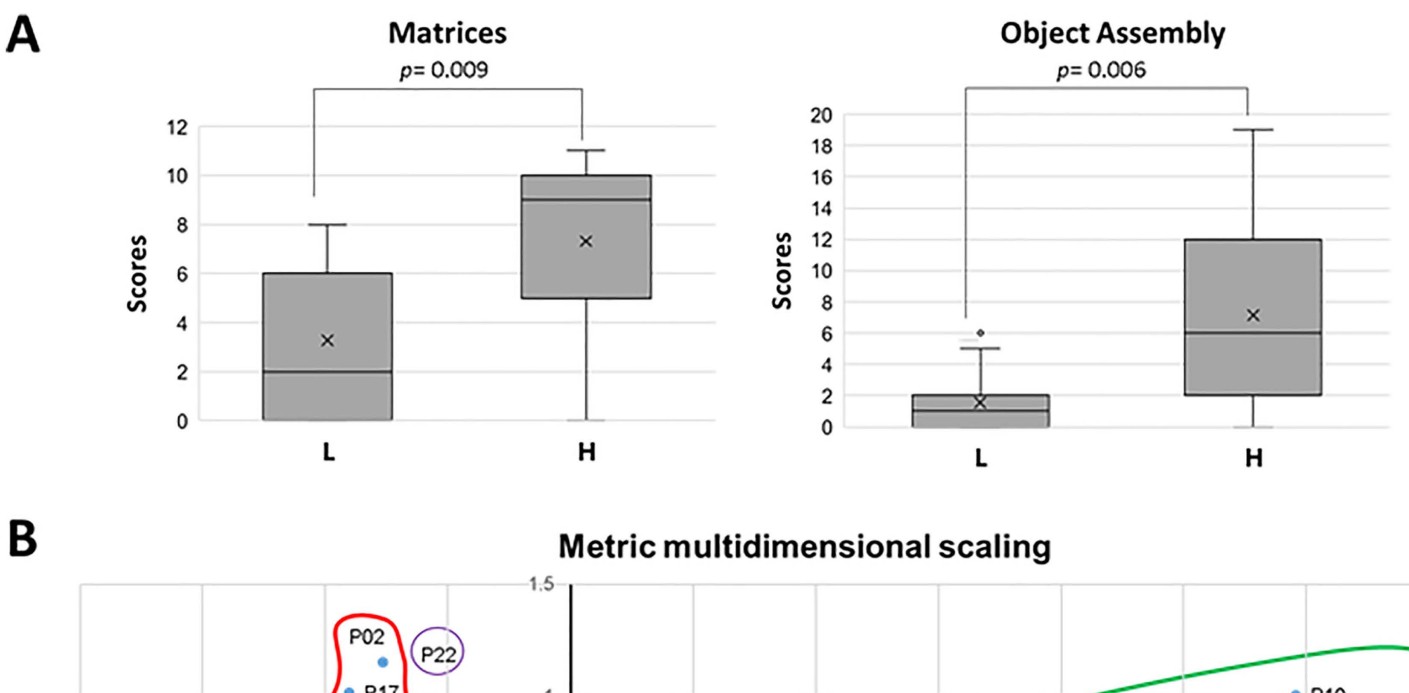

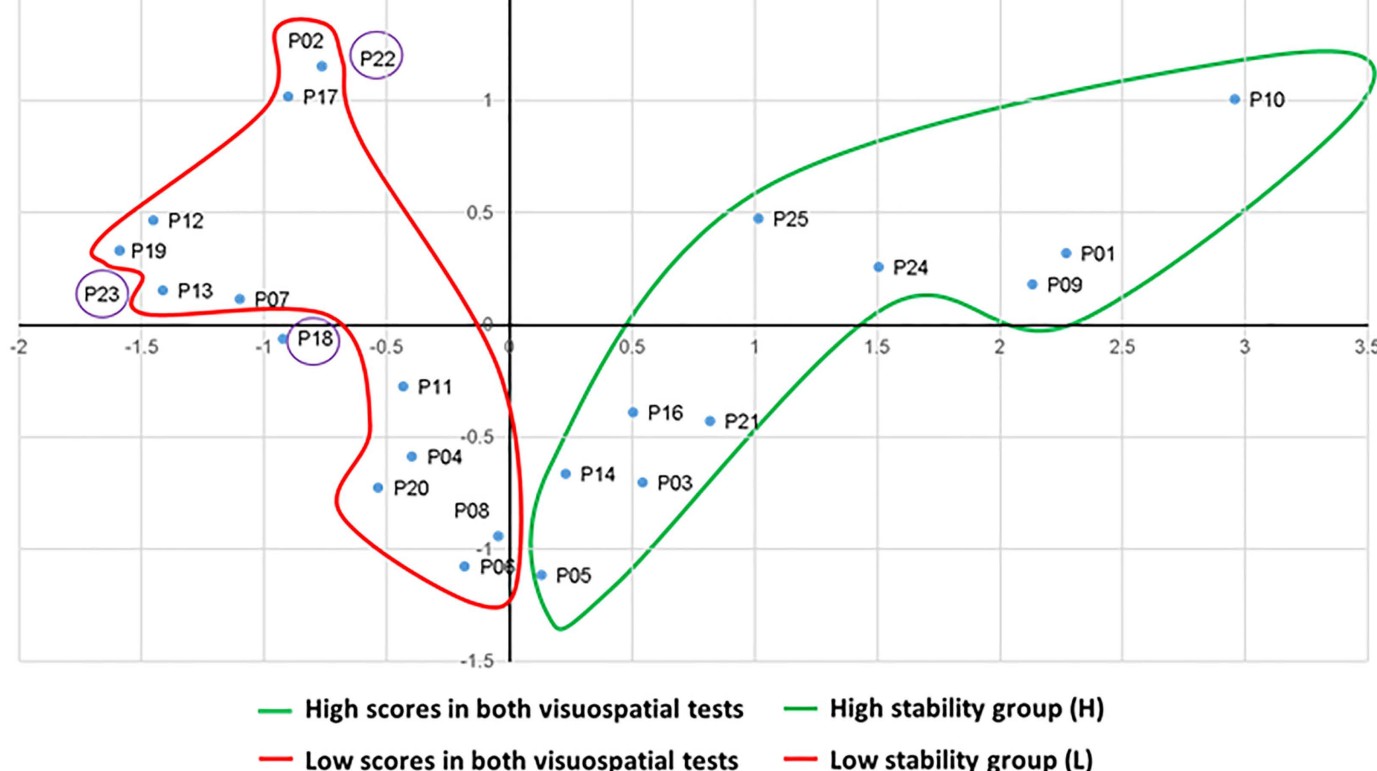

**Fig 5. Differences between the High stability (H) and Low stability (L) groups of patients in cognitive performance. (A)** Boxplots are shown of the performance of High stability (H patients) and Low stability (L patients) groups of patients in the Matrices test and in the Object Assembly test. Genotypes and group clustering of the patients are indicated in Tables 1 and S1. **(B)** Metric multidimensional analysis of High stability and Low stability groups of patients. The patients (P) located on the right-hand side of the graph have higher scores on both visuospatial cognitive tests (Matrices and Object Assembly tests), whereas the patients located on the left-hand side of the graph showed lower scores in both tests. The participants located at the top of the graph obtained better scores at the Object Assembly test compared to the Matrices test, whereas participants located at the bottom of the graph obtained higher scores at the Matrices test compared to the Object Assembly test. The patients that do not merge to the High (green) and Low (red) stability groups according to their cognitive scores are marked in purple. *RSQ* = 1.000; *S-stress* = 0.000.

stability group (_M_age = 5.32 years old), making logical to assume that the H group patients would be more autonomous for daily living tasks. In contrast, in the case of Matrices and Object Assembly cognitive tests, the cognitive domains assessed are not biased by age. Specifically, these two tests assess logical and spatial abilities, also known as fluid intelligence, which does not rely on learning experiences or accumulated knowledge. Together, our observations suggest the possibility of predicting some clinical outcomes of the disease on some CTNNB1 patients based on their specific _CTNNB1_ genetic alteration.

### Functional reconstitution of β-catenin pathogenic variants by proteasome inhibition

Next, we tested whether the steady-state expression of β-catenin variants was affected by the proteasome inhibitor MG-132 (Fig 6A). Transfected cells incubated in the presence of MG-132 displayed higher expression of both R474* and R661* variants, indicating the involvement of the proteasome in their degradation. We also performed transcriptional activity experiments of the R474* and R661* β-catenin variants upon cell treatment with MG-132 (Fig 6B). As shown, MG-132 cell treatment increased the transcriptional activity of β-catenin R474*, in accordance with its positive effect increasing R474* protein expression levels. Transcriptional activity of β-catenin wild type was also increased upon MG-132 cell treatment. These results indicate that inhibition of β-catenin protein degradation by the proteasome results in increased levels of β-catenin transcriptional activity.

### Functional reconstitution of β-catenin pathogenic variants by translational readthrough

Expression of full-length proteins from genes containing PTC in their coding regions can be facilitated by pharmacologically-induced translational readthrough, in a manner which is mainly dependent on the PTC identity and its nucleotide sequence context [35]. Therefore, we tested the inducible readthrough response of the β-catenin variants under study containing PTC. Transfected cells were incubated in the presence of the aminoglycoside readthrough inducer G418/geneticin, and expression of full-length β-catenin was monitored by immunoblot (Fig 7A). β-catenin R474* showed the best readthrough response to G418, followed by β-catenin R90*, whereas the rest of β-catenin PTC variants displayed very weak or none readthrough induction by G418. Both R474* and R90* are TGA PTC variants, which is consistent with the general rule of readthrough efficiency based on the identity of the PTC (TGA > TAG > TAA) [36,37]. The combination of

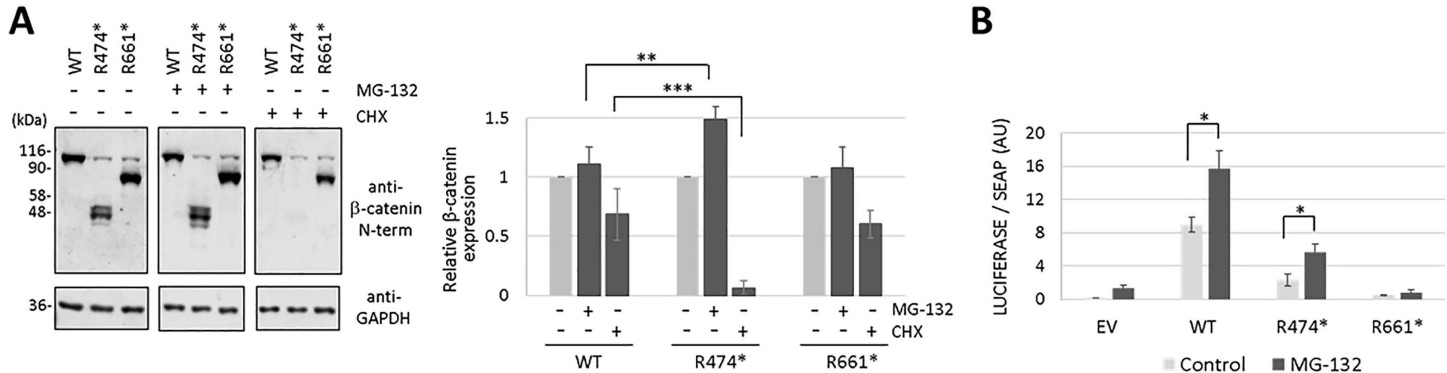

**Fig 6. Reconstitution of β-catenin function by proteasome inhibition. (A)** COS-7 cells were transfected with plasmids encoding the β-catenin variants (WT, wild type) and kept untreated (-) or incubated (+) in the presence of MG-132 (10 μM) or cycloheximide (CHX, 800 μg/ml) for 6 h. The left panel shows representative immunoblots using anti-β-catenin N-terminal or anti-GAPDH antibodies. The right panel shows the relative quantification of the bands from each variant, from at least two independent experiments. **, p = 0.0003; ***, p < 0.0001. **(B)** Transcriptional activity of β-catenin variants. COS-7 cells were transfected with plasmids encoding the indicated β-catenin variants (EV, empty vector; WT, wild type) and kept untreated (Control) or incubated in the presence of MG-132 (10 μM) for 6 h, as indicated. Cells were processed for TCF/LEF-driven luciferase activity, and luminescence is shown in arbitrary units (AU), from at least two independent experiments. *, p < 0.005.

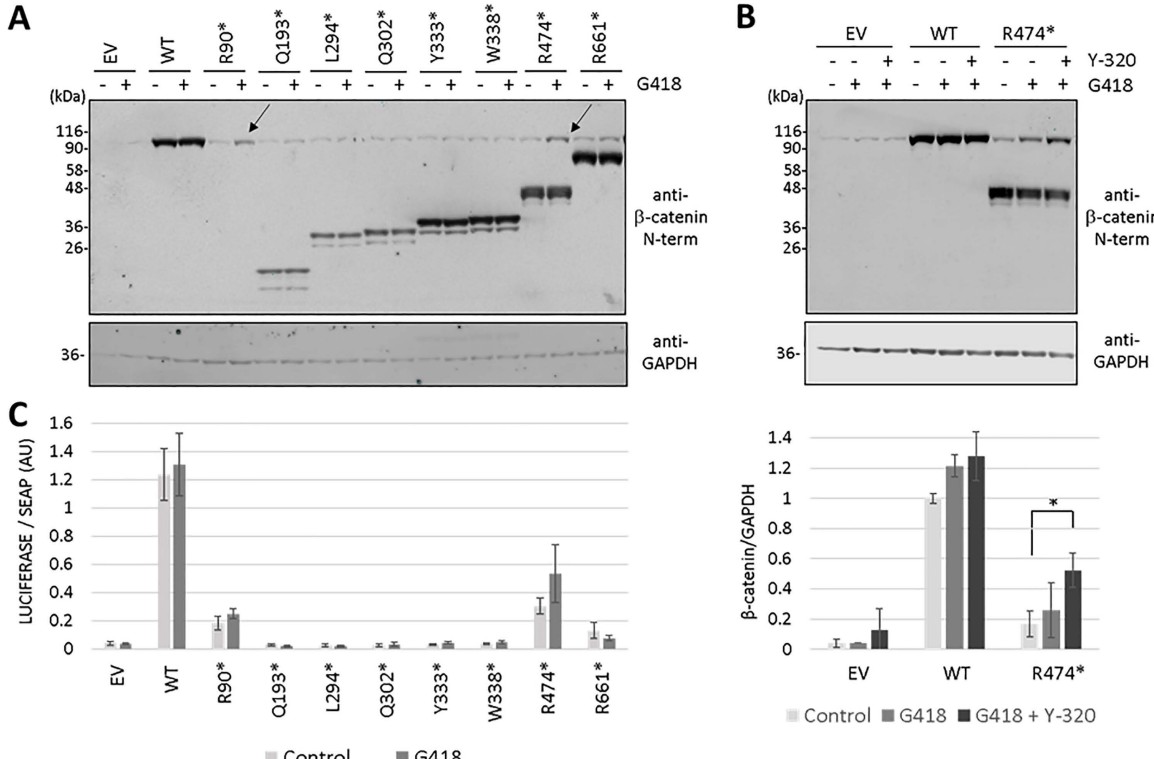

**Fig 7. Reconstitution of β-catenin function by translational readthrough. (A)** COS-7 cells were transfected with plasmids encoding the β-catenin variants (EV, empty vector; WT, wild type) and kept untreated (-) or incubated (+) in the presence of G418 (200 µg/ml) for 24 h. A representative immunoblot is shown using anti-β-catenin N-terminal or anti-GAPDH antibodies. **(B)** COS-7 cells were transfected with plasmids encoding the R474* β-catenin variant (EV, empty vector; WT, wild type) and kept untreated (-) or incubated (+) in the presence of G418 (200 µg/ml) and Y-320 (1 µM) for 24 h, as indicated. The top panel shows a representative immunoblot using anti-β-catenin N-terminal or anti-GAPDH antibodies. The bottom panel shows the relative quantification of the β-catenin full-length bands from each variant, from at least two independent experiments. *, p = 0.0133. **(C)** Transcriptional activity of β-catenin variants. COS-7 cells were transfected with plasmids encoding the indicated β-catenin variants (EV, empty vector; WT, wild type), and kept untreated (Control) or incubated in the presence of G418 (200 µg/ml) for 24 h, as indicated. Cells were processed for TCF/LEF-driven luciferase activity, and luminiscence is shown in arbitrary units (AU), from at least two independent experiments.

G418 and the protein synthesis inducer Y-320 further increased β-catenin R474* readthrough efficiency (Fig 7B). Finally, we tested the functional reconstitution of β-catenin R474* upon readthrough induction conditions. As shown, R474* variant transcriptional activity was moderately increased in the presence of G418 (Fig 7C).

These findings suggest that translational readthrough is feasible to reconstitute the expression and function of specific β-catenin PTC variants, which could have therapeutic potential for specific groups of CTNNB1 syndrome patients.

## Discussion

CTNNB1 syndrome is precisely defined at the genetic level by potential loss-of-function variants in the *CTNNB1* gene, encoding β-catenin [1]. However, CTNNB1 syndrome patients display a highly heterogeneous clinical phenotype, which covers a wide range of physical and behavioral manifestations [9]. This demands dedicated studies on the molecular and functional properties of β-catenin variants genetically identified on patients, and their potential correlations with patient phenotypes. We have here analysed at the molecular and functional level the group of β-catenin variants present in a comprehensive cohort of Spanish CTNNB1 syndrome patients, which have been characterized clinically and

neuropsychologically in detail [32,33]. Our cohort is mostly represented by β-catenin truncated variants generated by nonsense or frameshift variants in *CTNNB1*, which goes in line with the cohorts described in other studies [5–8,10–12]. All β-catenin truncated variants could be efficiently expressed *in vitro*, although with different relative expression levels. This suggests the possibility of the expression of β-catenin truncated proteins in patients, which needs to be confirmed by analysis of endogenous β-catenin from patient cells. The β-catenin variants from our cohort showed a complete or a partial loss of transcriptional activity. On the other hand, some *CTNNB1* missense variants associated to CTNNB1 syndrome display full transcriptional activity [30,38]. This suggests the potential pathogenic importance of both transcriptional and non-transcriptional β-catenin activities, whose examination warrants further studies.

Analysis of the relative expression of the variants from our cohort allowed us to distinguish between β-catenin proteoforms with larger deletions showing lower protein stability, and proteoforms with smaller deletions showing higher protein stability. We found statistically significant differences between high stability β-catenin variants compared to low stability in patient performance in visuospatial tests, which are suggestive of the implication of β-catenin in visuospatial memory, representing a possible deficit in the hippocampus due to non-optimal levels of β-catenin [39]. Some patients were outliers in this comparison, which could be explained by the high patient clinical heterogeneity found in CTNNB1 syndrome. Our results suggest that patients with high stability β-catenin variants would show better performance at visuospatial integration and object assembly tasks. This is in line with the proposal that *CTNNB1* truncating variants targeting the C-terminal portion of β-catenin could alter at lesser extent certain β-catenin functions [2]. In this regard, it has also been proposed that the position of PTC in the *DMD* gene in patients with Duchenne muscular dystrophy can predict the phenotype severity [40]. We did not find differences in other clinical variables evaluated in our phenotypic analysis, such as expressive and receptive language, executive functions, adaptive functioning and clinical manifestations [32,33]. These findings highlight the difficulties to establish genotype-phenotype correlations in CTNNB1 syndrome but suggest that specific clinical manifestations could be predicted on patients based on their genotype, which may be important for patient follow-up and the implementation of palliative therapies. For instance, the patient harboring the β-catenin variant Δ433–435, which displayed high protein stability and relatively high transcriptional activity, was among the patients from our cohort manifesting a milder to normal clinical phenotype in specific clinical assessments. Our analyses are based in a relatively small patient cohort, and further genotype-phenotype analysis is warranted with wider and independent cohorts of CTNNB1 syndrome patients.

We found that alternative translation initiation of β-catenin could be achieved in the presence of PTC at β-catenin N-terminal regions. This reflects nonsense variant-dependent reinitiation of translation, which has been reported for other proteins [41–44]. Our functional analyses of the truncated β-catenin pathogenic variants show the importance of both N-terminal and C-terminal regions of β-catenin for proper transcriptional activity, which is in agreement with previous reports using experimental N-terminal and C-terminal β-catenin truncations [45,46], as well as pathogenic C-terminal frameshift variants [31]. The N-terminal region of β-catenin is essential for regulation of β-catenin destruction complex assembly, β-catenin proteasome degradation, and β-catenin nuclear accumulation, whereas the β-catenin highly acidic C-terminal region has a major transcriptional activation function [18,47,48]. The N-terminal region of β-catenin binds to α-catenin, whereas the β-catenin armadillo-repeat core domain is involved in multiple protein-protein interactions, including binding to cadherins, axins and APC, among others [14]. An increase in AXIN1 binding to pathogenic C-terminal truncated β-catenin variants has been reported [31]. How the distinct pathogenic variants from CTNNB1 syndrome patients affect the multiple protein interactions involving β-catenin is mostly unexplored.

A major regulatory mechanism of β-catenin function in cells involves its proteasome-mediated degradation, which is triggered upon sequential phosphorylation of β-catenin N-terminus by CK1 and GSK3 [21,22]. The recent finding that GSK3α/β inhibition ameliorates the phenotype of a CTNNB1 syndrome mouse model gives hopes for efficacious therapeutic approaches based in restoring physiologic β-catenin levels in CTNNB1 syndrome patients [23]. Our results show that proteasome inhibition partially rescues the lack of transcriptional activity of some CTNNB1 syndrome β-catenin variants; however, since MG-132 broadly inhibits global protein degradation, off-target and toxic effects can be produced that

may prevent the feasibility of this compound for clinical use. Liu et al. also reported the restoration, upon proteasome inhibition or GSK3β inhibition, of β-catenin protein levels and function in pathogenic β-catenin C-terminal truncated variants [31]. This suggests that appropriate doses and combinations of inhibitors of β-catenin degradation could be beneficial for CTNNB1 syndrome patients. In this regard, the GSK3β inhibitor tideglusib and the proteasome inhibitor bortezomib have shown safety and therapeutic efficacy in advanced clinical trials in children with congenital myotonic dystrophy or with acute leukemias and lymphomas, respectively [49,50].

CTNNB1 nonsense and frameshift variants generating PTC are frequent in CTNNB1 syndrome patients, and mRNA harboring PTC are targeted for degradation by the RNA turnover pathway of NMD [25,51]. However, the efficiency of NMD is variable depending on intrinsic and extrinsic factors, including PTC location, nucleotide context, and cell type. For instance, mRNA harbouring PTC 50–55 nucleotides upstream of the last exon-exon junction escape NMD degradation [52,53]. This makes possible, in some cases, the persistence on cells of stable mRNA containing PTC [54,55]. In vitro studies with patient-derived cells will be necessary to verify this hypothesis.

During protein biosynthesis, translational readthrough incorporates an amino acid encoded by a near-cognate codon in the place occupied by a PTC, a process that is highly dependent on the PTC identity and its nucleotide context [28,29]. We have evaluated the efficacy of full-length protein reconstitution by induced translational readthrough on the eight β-catenin variants from our cohort that are targeted by nonsense variants. We have found that expression of full-length active β-catenin can be induced by the antibiotic G418 and by the protein synthesis stimulator Y-320 in two β-catenin variants, R90* and R474*, which is in line with the overall higher readthrough efficiency observed for TGA PTC in other genes [36,37,56]. In addition, the nucleotide context around the PTC is also important for the efficiency of induced readthrough (reviewed in [57]. A major limitation of translational readthrough is the toxicity associated to many readthrough inducers, including the aminoglycoside G418, making a priority in the field the identification and clinical validation of non-toxic compounds displaying optimal readthrough activity [58]. Further analysis is required to test the efficacy of non-toxic readthrough-inducing compounds on reconstitution of β-catenin PTC variants. In addition, incorporation during readthrough of non-wild type amino acids in the PTC position is possible, making necessary the functional analysis of the reconstituted protein, as illustrated here. On the other hand, a major advantage of translational readthrough is the stabilization of the PTC mRNA, avoiding its degradation by NMD [59,60]. Our results show that β-catenin expression and function can be partially restored from specific CTNNB1 nonsense variants, although validation of clinically feasible readthrough-inducing compounds is necessary. We suggest that specific groups of CTNNB1 syndrome patients could get benefit from non-toxic pharmacologic readthrough inducers.

## Materials and methods

### Cell Culture, transfections, and reagents

Simian kidney COS-7 cells were grown at 37°C, 5% $CO_2$ in DMEM containing high glucose supplemented with 5% heat-inactivated fetal bovine serum (FBS), 1 mM L-glutamine, 100 U/ml penicillin, and 0.1 mg/ml streptomycin. Cells were transfected using GenJet reagent (SignaGen Laboratories) according to the manufacturer instructions. Transfected cells were cultured for 24 h, and were incubated for additional 24 h before harvesting. For protein stability assays, cells were treated with cycloheximide (CHX, 800 µg/mL) (Merck Sigma-Aldrich) or MG-132 (10 µM) (Merck Sigma-Aldrich) for the indicated time, as previously described [61]. For readthrough induction experiments, cells were incubated in the presence of G418 (200 µg/mL) (Merck Sigma Aldrich) or Y-320 (1 µM) (MedChemExpress) for 24 h [62].

### Plasmids and mutagenesis

The mammalian expression plasmid pRK5 CTNNB1/β-catenin (human sequence) has been previously described [34,63]. For production of recombinant β-catenin in E. coli, CTNNB1 cDNA was subcloned in a modified pTG-A20 plasmid [64],

conferring an N-terminal (His)$_{10}$-ZZ tag. Mutagenesis was performed by oligonucleotide site-directed PCR mutagenesis as described [65], and variants were confirmed by DNA sequencing. Nucleotide and amino acid numbering for human CTNN-B1/β-catenin cDNA used in the study is according to entries NM_001904 and NP_001895, respectively.

## Immunoblot and antibodies

Whole cell protein extracts from COS-7 cells transfected with plasmids encoding β-catenin wild type or variants were prepared by cell lysis in ice-cold M-PER lysis buffer (ThermoFisher Scientific) supplemented with PhosSTOP phosphatase inhibitor and cOmplete protease inhibitor cocktails (Roche). Cell lysates were centrifuged at 15200 g for 10 min and the supernatant was collected. Proteins (50–100 µg) were resolved in 10% SDS-PAGE under reducing conditions and transferred to PVDF membranes. Immunoblots were performed using anti-β-catenin N-terminal mAb (Non-phospho (Active) β-Catenin (Ser45) (D2U8Y) XP Rabbit mAb #19807, Cell Signaling), anti-β-catenin C-terminal mAb (β-Catenin (D10A8) XP Rabbit mAb #8480, Cell Signaling), or anti-GAPDH antibody (GAPDH (6C5) Mouse mAb sc-32233, Santa Cruz Biotechnology), diluted in immunoblot blocking buffer [dilution 1:1 in PBS of *Odyssey Blocking Buffer* (*OBB buffer,* LI-COR Biosciences)], followed by IRDye-conjugated anti-mouse antibody (LI-COR Biosciences). For protein stability experiments, protein bands were quantified using an Image Studio Software with Odyssey CLx Imaging System (LI-COR Biosciences).

## Circular dichroism spectroscopy

Recombinant CTNNB1 variants were purified by Ni-NTA chromatography. CD spectra were recorded with a Jasco 720 spectropolarimeter, using a quartz cuvette of 2 mm path length, in buffer 20 mM Tris/HCl, pH 8.0, 50 mM NaCl and at 20ºC. Protein concentration was 1 µM. Mean residue ellipticity values were calculated using the expression $\Theta = \varepsilon/10cln$, where $\varepsilon$ is the ellipticity (millidegrees), $c$ the protein concentration (mol/L), $l$ is the path length (cm), and $n$ is the number of peptide bonds of the protein.

## Luciferase reporter assay

COS-7 cells were plated in 96-well plates and transfected on the following day as indicated above and as reported in [34]. Triplicates of each variant or condition were analysed for intracellular luciferase activity 48 h post-transfection according to Luciferase Reporter Assay kit (Promega). Culture media was harvested for measuring secreted alkaline phosphatase (SEAP) using Secrete-Pair Dual Luminescence Assay kit (GeneCopoeia). SEAP values were used to normalize for differences in transfection efficiency. Luminescence was measured using TECAN plate reader.

## Phenotypic and statistical analysis

Receptive and expressive vocabulary tests, visuospatial integration and object assembly performance, immediate memory and executive functioning were assessed. Also, adaptive and motor functioning, sleep problems, and communication and eating efficacy were assessed through standardized tests. The protocol used has been described in detail [32,33]. For genotype-phenotype analyses, the statistical program used was SPSS (Statistical Package for Social Sciences) version 28.0. Descriptive statistics were performed for all measures and, to compare different variables, raw scores were converted into *Z* scores. Non-parametric analyses were carried out due to sample size. To compare results between groups we used a Mann-Whitney *U* test and a Metric multidimensional analysis to establish similarities and differences observed between the subjects. A two-way ANOVA with Dunnet's post-hoc test (Fig 6) or a two-way ANOVA followed by Tukey's post-hoc test (Fig 7) was conducted to assess the effects of the variants and treatments on protein expression. To evaluate statistical significance in the luciferase experiments, the two-tailed student *t* test was used.

## Supporting information

**S1 Fig. Relative expression levels of the β-catenin variants analysed in this study.** COS-7 cells were transfected with pRK5 mammalian expression plasmids encoding the indicated β-catenin variants and processed for immunoblot as in Fig 2, using anti-β-catenin N-terminal or C-terminal antibodies. Protein bands were quantified and represented with respect to β-catenin wild type. Data are shown as relative expression ± SD, from at least three independent experiments.
(PNG)

**S1 Table. Patient classification by genotype groups.**
(PDF)

**S2 Table. Clinical, psychological and cognitive domains differences between patient groups.**
(PDF)

**S1 Data. Data that underlies this paper.**
(XLSX)

## Acknowledgments

We are grateful to the Asociación CTNNB1 España (https://asociacionctnnb1.org) and all the CTNNB1 syndrome families for their continuous support. We thank personnel from the Genetics-Genomic Core facility, and all technical and administrative personnel from Biobizkaia Health Research Institute, for their expert assistance.

## Author contributions

**Conceptualization:** Caroline E Nunes-Xavier, Mercè Pallarès-Sastre, Ana Rodríguez-Ramos, Sonia Bañuelos, Fabio Cavaliere, Maitane García, Imanol Amayra, Rafael Pulido.

**Data curation:** Caroline E Nunes-Xavier, Mercè Pallarès-Sastre, Ana Rodríguez-Ramos, Sonia Bañuelos, Rafael Pulido.

**Formal analysis:** Caroline E Nunes-Xavier, Mercè Pallarès-Sastre, Ana Rodríguez-Ramos, Sonia Bañuelos, Rafael Pulido.

**Funding acquisition:** Caroline E Nunes-Xavier, Rafael Pulido.

**Investigation:** Caroline E Nunes-Xavier, Mercè Pallarès-Sastre, Ana Rodríguez-Ramos, Sonia Bañuelos, Irune Cortajarena, Rafael Pulido.

**Methodology:** Caroline E Nunes-Xavier, Mercè Pallarès-Sastre, Ana Rodríguez-Ramos, Sonia Bañuelos, Irune Cortajarena, Rafael Pulido.

**Resources:** Caroline E Nunes-Xavier, Cynthia Ruiz-Espinoza, Isabel Llano-Rivas.

**Supervision:** Caroline E Nunes-Xavier, Mercè Pallarès-Sastre, Ana Rodríguez-Ramos, Sonia Bañuelos, Maitane García, Imanol Amayra, Rafael Pulido.

**Validation:** Caroline E Nunes-Xavier, Mercè Pallarès-Sastre, Ana Rodríguez-Ramos, Sonia Bañuelos, Fabio Cavaliere, Maitane García, Rafael Pulido.

**Writing – original draft:** Caroline E Nunes-Xavier, Rafael Pulido.

**Writing – review & editing:** Caroline E Nunes-Xavier, Mercè Pallarès-Sastre, Ana Rodríguez-Ramos, Sonia Bañuelos, Irune Cortajarena, Fabio Cavaliere, Cynthia Ruiz-Espinoza, Isabel Llano-Rivas, Maitane García, Imanol Amayra, Rafael Pulido.

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
