## [Decision Letter · Decision Letter 0]

15 May 2025

PGENETICS-D-25-00398

Genotype-phenotype characterization and functional reconstitution of pathological ß-catenin variants from CTNNB1 syndrome patients

PLOS Genetics

Dear Dr. Pulido,

Thank you for submitting your manuscript to PLOS Genetics. After careful consideration, we feel that it has merit but does not fully meet PLOS Genetics's publication criteria as it currently stands. Therefore, we invite you to submit a revised version of the manuscript that addresses the points raised during the review process.

Please submit your revised manuscript within 60 days Jul 14 2025 11:59PM. If you will need more time than this to complete your revisions, please reply to this message or contact the journal office at plosgenetics@plos.org. Please include the following items when submitting your revised manuscript:

We look forward to receiving your revised manuscript.

Kind regards,

Kevin E. Glinton, MD, PHD

Academic Editor

PLOS Genetics

Hua Tang

Section Editor

PLOS Genetics

Aimée Dudley

Editor-in-Chief

PLOS Genetics

Anne Goriely

Editor-in-Chief

PLOS Genetics

**Journal Requirements:**

At this stage, the following Authors/Authors require contributions: Caroline E Nunes-Xavier, Mercè Pallarès-Sastre, Sonia Bañuelos, Irune Cortajarena, Fabio Cavaliere, Cynthia Ruiz-Espinoza, Isabel Llano-Rivas, Maitane García, Imanol Amayra, and Rafael Pulido. Please ensure that the full contributions of each author are acknowledged in the "Add/Edit/Remove Authors" section of our submission form.

The list of CRediT author contributions may be found here: https://journals.plos.org/plosgenetics/s/authorship#loc-author-contributions

https://journals.plos.org/plosgenetics/s/submission-guidelines#loc-parts-of-a-submission

4) We do not publish any copyright or trademark symbols that usually accompany proprietary names, eg ©,  ®, or TM  (e.g. next to drug or reagent names). Therefore please remove all instances of trademark/copyright symbols throughout the text, including:

- ® on page: 19

- TM on pages: 18, and 19.

5) Please upload all main figures as separate Figure files in .tif or .eps format. For more information about how to convert and format your figure files please see our guidelines: 

6) We have noticed that you have uploaded Supporting Information files, but you have not included a list of legends. Please add a full list of legends for your Supporting Information files after the references list.

7) We note that your Data Availability Statement is currently as follows: "All relevant data are within the paper". Please confirm at this time whether or not your submission contains all raw data required to replicate the results of your study. Authors must share the “minimal data set” for their submission. PLOS defines the minimal data set to consist of the data required to replicate all study findings reported in the article, as well as related metadata and methods (https://journals.plos.org/plosone/s/data-availability#loc-minimal-data-set-definition).

8) Thank you for stating "The founders did not play any role in the study design, data collection and analysis, decision to publish, or preparation of the manuscript." Please amend your detailed Financial Disclosure statement. This is published with the article. It must therefore be completed in full sentences and contain the exact wording you wish to be published. Please state: "The funders had no role in study design, data collection and analysis, decision to publish, or preparation of the manuscript."

9) Please ensure that the funders and grant numbers match between the Financial Disclosure field and the Funding Information tab in your submission form. Note that the funders must be provided in the same order in both places as well. Currently, " CEN-X is the recipient of a Miguel Servet Research Contract from Instituto de Salud Carlos III (CP20/00008, Spain and co-funded by European Union). MP is funded by Ministry of Sciences, Innovation and Universities of Spain (FPU22/00391). RP is funded by Ikerbasque, The Basque Foundation for Science, Spain" are missing from the Funding Information tab.

**Reviewers' comments:**

Reviewer's Responses to Questions

Reviewer #1: Nunes-Xavier and colleagues performed an interesting study on the residual expression levels and transcriptional activities of β-catenin proteoforms generated from premature termination codons found in patients carrying germline CTNNB1 mutations. They attempt to correlate these findings with the patients’ phenotypes.

The study addresses an unresolved question about the heterogeneity of CTNNB1 syndrome and is well designed in its experimental section.

Nevertheless, my main concern relates to the accuracy and methodological rigor of the genotype-phenotype correlations. The authors report significant differences in only two items (out of six) of the Wechsler Nonverbal Scale of Ability between two groups of patients clustered according to protein stability data.

It is unclear whether the authors investigated differences between these two groups in other adaptive and neuropsychological assessments, and if so, whether any significant findings survived correction for multiple comparisons. The phenotypic protocol is only referenced via previous publications from the same group. Additionally, the lack of significant differences in other clinical variables assessed—such as expressive and receptive language, executive functions, adaptive functioning, and clinical manifestations—is only vaguely mentioned with reference to other “under review” papers.

For the sake of scientific accuracy, the phenotypic features analyzed and compared between groups should be explicitly listed, and the statistical methods used should be reported.

Furthermore, there appears to be no clear correlation between protein stability and residual transcriptional activity, with many variants classified as “highly stable” exhibiting apparently poor transcriptional activity. Why did the authors not explore whether residual transcriptional activity correlates with phenotype? It is unclear why the authors grouped patients solely based on protein stability, given that their own data (e.g., Figure 7) suggest that the reduced degradation of transcriptionally inactive proteoforms does not enhance their transcriptional function.

In summary, to address these concerns, the authors should:

• Provide a detailed phenotypic classification of the cohort across all domains (motor, cognitive, adaptive);

• Justify their rationale for clustering variants based on protein stability (or consider re-clustering based on residual transcriptional activity, which may offer better external validity);

• Clearly report the statistical methods used in the genotype-phenotype analysis.

Minor revisions:

FIGURE 3: A and B should be added to the figure.

Reviewer #2: Authors perform an in vitro analysis of truncated pathogenic CTNNB1 variants cloned into a mammalian expression vector from a full length cDNA construct. Truncated proteins show a varying degree of stability or inefficiency of translation compared with the wild type protein and also varying degrees of effectiveness in mediating transcriptional activation of a luciferase reporter. A modest phenotype to genotype correlation is reported and an assessment of feasibility of transcriptional read-through using aminoglycosides as treatment.

Major comments / questions

1. Throughout this paper there is minimal consideration given to the effects of non-sense mediated decay (NMD) elicited at the transcript level in the context of patient cell lines. If the message is degraded by NMD the relative stability of the predicted protein as determined by ectopic expression from a cDNA sequence is inconsequential.

-Do the authors have access to patient cell lines or RNA extracted from blood where measurements of NMD could be made?

-The aminoglycoside (or alternative PTC read through approach) and protein stabilisation experiments would have much greater impact if performed on patient derived cell lines.

2. The N-terminally truncated forms of the protein that are formed from re-initiation of translation do not have the b-TrCP binding site. Even though the efficiency of translation re-initiation is low the overall stability of these proteins should no longer be dictated by the destruction complex.

-The results of the cycloheximide block are at odds with this where there should be an expectation that CTNNB1 full-length protein should degrade rapidly in the absence of translation? It would be better to see these N-terminal truncations assessed by CHX chase as is shown in Fig 5B for p.Arg474Ter and p.Arg661Ter and / or inclusion of a p.Ser33Ala full length CTNNB1 which should remain stable for comparison to show that the destruction complex is active in the culture system in use.

-Are there phenotypic differences specific to these patients compared to others with PTC variants that do not re-initiate?

-Are these transcripts subject to NMD in patient cells or if unavailable when these variants are knocked into the CTNNB1 gene in a cell line?

3. Genotype to phenotype correlations made between high vs. low stability patients are very subtle and there are also outliers in the MDS plot.

-What other covariates were considered especially e.g. biological sex, socio-economic status, underlying polygenic risk for low educational attainment?

-What is the significance for the patient for performing better or worse in these specific domains given many other clinical outcomes are equivalent?

5. As the authors note aminoglycosides are not a feasible treatment however other compounds available such as PTC124 (Ataluren) could be used and is easily obtainable and is more effective. Why was this approach excluded?

6. In the discussion the authors discuss the efficacy of restoration of CTNNB1 by GSK3B. In the KO model, the mutant allele is completely non-functional and does not express Ctnnb1 so GSK3B inhibition is only protecting the wild type allele from degradation by the destruction complex not restoring the mutant allele. Looking only at transcriptional activity also disregards the role of CTNNB1 in cell-cell adhesion where C-terminal truncations may not form strong interactions with cadherins . There are some missense variants implicated in CTNNB1 syndrome that have no effect on the topflash assay compared to wild type CTNNB1, e.g. the recurrent p.Gly557Arg and also the Bfc mutation in mice is slightly gain of function in this assay. The risks of expressing truncated proteins that the NMD and unfolded protein responses have evolved to prevent should be discussed.

Minor:

7. Throughout: Gene symbols for all species i.e. CTNNB1/Ctnnb1 should be in italics.

8. Line 64-65: Check grammar: "The syndrome is characterized by a generalized delay in children development and shows a great phenotypic variety."

9. The use of X to denote a termination codon is incorrect. In the IUPAC code X means a non-specific amino acid. The correct single character code for a termination codon is * see https://hgvs-nomenclature.org/stable/recommendations/protein/substitution/ and https://iupac.qmul.ac.uk/AminoAcid/AA1n2.html

10. Terminology switches between mutation and variant. Suggest to use the term variant for this work because it is more appropriate for human related literature.

11. The pRK5 plasmid is referred as described in ref 55 but that reference directs to an earlier one with the quote "Mammalian expression plasmid, pRK5 β-catenin, has been previously reported" Fuchs M, Müller T, Lerch MM, Ullrich A. J Biol Chem. 1996 Jul 12;271(28):16712-9.

Reviewer #3: Nunes-Xavier et al. examined the expression and activity of both N-terminal and C-terminal β-catenin in a series of clinically identified CTNNB1 variants. Their results revealed that the variants produced truncated β-catenin proteins with varying degrees of stability. Notably, longer truncations were associated with increased protein stability, which in turn correlated with improved cognitive outcomes in patients. Intriguingly, some of the truncated variants appeared to undergo partial functional reconstitution via translational readthrough, suggesting a possible therapeutic avenue for CTNNB1 syndrome. Overall, the manuscript is well-written, methodologically sound, and presents important insights into β-catenin biology. However, several minor issues should be addressed to improve the clarity and rigor of the study:

1. The analyses presented in Figures 5B, 7A, and 8B may not be optimal. Given the multiple conditions and variables tested, a two-way ANOVA would be more appropriate to evaluate potential interaction effects.

2. Statistical significance should be explicitly indicated within the figures, rather than described solely in the legends, as seen in Figure 6A.

3. The panels in Figure 3 should be clearly labeled as A and B to enhance interpretability.

4. Figures 2 and 3 appear to be conceptually linked, as Figure 3A summarizes data from Figure 2. Merging these into a single figure could improve logical flow and reduce redundancy.

5. While Figures 2 and 3 illustrate the presence of β-catenin isoforms with distinct molecular weights across variants, it would be informative to also include quantitative data on expression levels for each variant.

6. The x-axis in the right panel of Figure 7A presents treatment groups, whereas other figures use gene variants. To facilitate comparison across figures, a uniform format for data presentation is recommended.

7. The rationale for analyzing only N-terminal β-catenin in Figure 5 should be clarified. Was C-terminal detection not informative?

8. Is G418 expected to induce translational readthrough universally across all PTCs in the tested variants?

9. The study utilizes CHX, MG-132, and Y-320 to modulate β-catenin expression. It should be clarified whether these reagents act specifically on β-catenin or exert broader effects on global protein synthesis and degradation. If they are nonspecific, a discussion of potential off-target or compensatory mechanisms is warranted.

10. The inhibitory effect of CHX on β-catenin expression is evident in Figure 5A but appears absent for the R661 variant in Figure 7A. This inconsistency should be addressed.

11. Although G418 promotes β-catenin upregulation, this effect is observed primarily in the R474X variant. The manuscript would benefit from a more detailed discussion on why G418’s efficacy is limited to certain variants and what factors might underlie this selectivity.

12. While MG-132, Y-320, and G418 show potential for enhancing β-catenin levels in specific variants, the translational potential of these compounds requires further discussion, especially in terms of specificity, toxicity, and feasibility for clinical use.

**Have all data underlying the figures and results presented in the manuscript been provided?**

Reviewer #1: Yes

Reviewer #2: **No: ** All graphs do not show individual data points.

Reviewer #3: Yes

PLOS authors have the option to publish the peer review history of their article (what does this mean? ). If published, this will include your full peer review and any attached files.

**Do you want your identity to be public for this peer review?** For information about this choice, including consent withdrawal, please see our Privacy Policy .

Reviewer #1: No

Reviewer #2: No

Reviewer #3: **Yes: ** Tao Tan

**Figure resubmission:**
---

## [Decision Letter · Decision Letter 1]

3 Oct 2025

Dear Dr Pulido,

We are pleased to inform you that your manuscript entitled "Genotype-phenotype characterization and functional reconstitution of pathogenic ß-catenin variants from CTNNB1 syndrome patients" has been editorially accepted for publication in PLOS Genetics. Congratulations!

Yours sincerely,

Kevin E. Glinton, MD, PHD

Academic Editor

PLOS Genetics

Hua Tang

Section Editor

PLOS Genetics

Aimée Dudley

Editor-in-Chief

PLOS Genetics

Anne Goriely

Editor-in-Chief

PLOS Genetics

BlueSky: @plos.bsky.social

Comments from the reviewers (if applicable):

Reviewer's Responses to Questions

**Comments to the Authors:**

Reviewer #2: I thank the authors for considering my comments. The paper is improved in my view by taking a more conservative approach to the findings and placing them in context of the limitations of the model used. I have only one amendment which is important to address.

NMD escape is only when the PTC is < 50-55 nucleotides from the final exon-exon junction in the gene (this catches many people out; see Lindeboom et al. for an example). The rule is poorly worded "translation termination at least 50 nt upstream of an exon junction triggers NMD" many assume that refers only to the most adjacent exon junction but actually it means any and all downstream exon junctions. All of the variants in this study will generate transcripts that have exon junction complexes downstream of the PTC > 50-55 nt and will all be subject to NMD, except those that reinitiate translation and are likely escape NMD that way.

- Table 1 legend and the discussion (line 483) require minor amendment to address this.

Congratulations, this is a great example of well presented molecular biology.

1. Lindeboom RGH, Supek F, Lehner B. The rules and impact of nonsense-mediated mRNA decay in human cancers. Nat Genet. 2016 Oct;48(10):1112–8.

Reviewer #3: Most of my concerns have been addressed, and the overall quality of the manuscript has improved. However, I noticed that the data in the revised left panel of Figure 6B appears to differ from the original graph (previously shown as the left panel of Figure 7B). This discrepancy should be carefully re-checked and validated to ensure consistency and accuracy.

**Have all data underlying the figures and results presented in the manuscript been provided?**

Reviewer #2: Yes

Reviewer #3: Yes

PLOS authors have the option to publish the peer review history of their article (what does this mean? ). If published, this will include your full peer review and any attached files.

**Do you want your identity to be public for this peer review?** For information about this choice, including consent withdrawal, please see our Privacy Policy .

Reviewer #2: No

Reviewer #3: **Yes: ** Tao Tan

**Data Deposition**

http://datadryad.org/submit?journalID=pgenetics&manu=PGENETICS-D-25-00398R1

**Press Queries**

---

## [Editor Report · Acceptance letter]

PGENETICS-D-25-00398R1

Genotype-phenotype characterization and functional reconstitution of pathogenic ß-catenin variants from CTNNB1 syndrome patients

Dear Dr Pulido,

We are pleased to inform you that your manuscript entitled "Genotype-phenotype characterization and functional reconstitution of pathogenic ß-catenin variants from CTNNB1 syndrome patients" has been formally accepted for publication in PLOS Genetics! Your manuscript is now with our production department and you will be notified of the publication date in due course.

With kind regards,

Anita Estes

PLOS Genetics

On behalf of:
